# In Vitro Activity of Ozone/Oxygen Gaseous Mixture against a Caprine Herpesvirus Type 1 Strain Isolated from a Goat with Vaginitis

**DOI:** 10.3390/ani13121920

**Published:** 2023-06-08

**Authors:** Edoardo Lillo, Francesco Pellegrini, Annalisa Rizzo, Gianvito Lanave, Claudia Zizzadoro, Vincenzo Cicirelli, Cristiana Catella, Michele Losurdo, Vito Martella, Maria Tempesta, Michele Camero

**Affiliations:** 1Department of Veterinary Medicine, University of Bari Aldo Moro, S.P. per Casamassima km. 3, 70010 Valenzano, BA, Italy; edoardo.lillo@uniba.it (E.L.); francesco.pellegrini@uniba.it (F.P.); gianvito.lanave@uniba.it (G.L.); claudia.zizzadoro@uniba.it (C.Z.); cristiana.catella@uniba.it (C.C.); michele.camero@uniba.it (M.C.); 2Department of Prevention of Animal Health and Welfare, Local Health Authority of Matera, Via Montescaglioso, 75100 Matera, BA, Italy

**Keywords:** ozone, caprine herpesvirus 1, in vitro virucidal activity, antiviral activity

## Abstract

**Simple Summary:**

*Alphaherpesviruses* cause genital lesions in both animals and humans. Ozone (O_3_) has a strong virucidal action on enveloped and naked viruses. The aim of this study was to test the in vitro virucidal and antiviral activity of an ozone/oxygen (O_3_/O_2_) gaseous mixture against caprine herpesvirus type 1 (CpHV-1). To test the virucidal activity, the virus was exposed to different concentrations (20 and 50 μg/mL) of the gaseous mixture at different time points, and a decrease in the viral titer by up to 2.0 log10 TCID_50_/50 μL was observed. To test the antiviral activity, the virus was exposed to different non-cytotoxic concentrations of the gaseous mixture. When MDBK cell monolayers were treated with the gas mixture after infection with CpHV-1 at a concentration of 50 μg/mL, significant antiviral activity was observed with a decrease in viral titer of 2.0 log10 TCID_50_/50 μL. These findings aid future studies aimed at assessing if topical treatment of genital herpes lesions in vivo with O_3_/O_2_ gaseous mixture could be a valid and safe therapeutic option in an animal model, with possible translational applications in the therapy of human herpes simplex virus type 2 (HSV-2), which shares several biological similarities with CpHV-1.

**Abstract:**

*Alphaherpesviruses* cause genital lesions and reproductive failure in both humans and animals. Their control is mainly based on prevention using hygienic prophylactic measures due to the absence of vaccines and limitations of antiviral drug therapy. Ozone is an oxidating gas showing a strong microbicidal activity on bacteria, fungi, viruses, and protozoa. The present study assessed the in vitro virucidal and antiviral activity of ozone against caprine herpesvirus type 1 (CpHV-1). The virucidal activity of a gaseous mixture containing O_3_ at 20 and 50 μg/mL was assessed against the virus at different contact times (30 s, 60 s, 90 s, 120 s, 180 s, and 300 s). Antiviral activity of a gaseous mixture containing O_3_ at 20 and 50 μg/mL was evaluated against the virus after 30 s and 60 s. Ozone displayed significant virucidal activity when used at all the tested concentrations whilst significant antiviral activity was observed using ozone at 50 μg/mL. The gaseous mixture, tested in the present study, showed virucidal and antiviral activity against CpHV-1 in a dose- and time contact-dependent fashion. Ozone therapy could be evaluated in vivo for the treatment of CpHV-1-induced genital lesions in goats using topical applications.

## 1. Introduction

Viral infections of the reproductive system are endemic in mammals and have negative repercussions on sexual and reproductive performances. Among them, the *Alphaherpesviruses* (family *Herpesviridae*, subfamily *Alphaherpesvirinae*) cause genital lesions and abortus in both humans and animals [1,2]. *Alphaherpesviruses* are large, enveloped DNA viruses characterized by rapid, lytic growth cycles [3]. Some herpesviruses infect the genital tract and subsequently establish a lifelong latent infection in the lumbosacral sensory ganglia that can be recurrently reactivated by stress, immunosuppression, or hormonal changes [4].

In humans, herpes simplex virus type 2 (HSV-2) is a major cause of genital infection, inducing painful genital ulcers, with 13% of the population aged 15–49 years being infected [2]. HSV-2 mainly causes genital herpes, which is the most common sexually transmitted ulcerative disease, and is considered a global health problem [5].

The control of HSV-2 is mainly based on prevention (through information and education) and on the use of viral DNA polymerase inhibitors [6]. These molecules can accelerate symptom resolution and lesion healing, but they cannot eradicate latent HSV infection and can induce drug resistance [7]. Resistance to antiviral drugs is a major problem in the fight against contagious diseases, such as influenza and hepatitis. The impact of resistance to antivirals can be important and fatal as it can affect drastically the effectiveness of therapy. This has driven the research to find alternative therapies.

*Alphaherpesviruses* also cause reproductive failure in farm animals and economic loss for the livestock industry [1]. Caprine herpesvirus type 1 (CpHV-1) is a widespread virus in goat herds and causes vulvovaginitis, balanoposthitis, infertility, abortions, and stillbirth [8]. Abortions associated with CpHV-1 occur during the second half of pregnancy and can be reproduced experimentally through intranasal and intravenous inoculation of pregnant goats [9]. CpHV-1 causes latent infections but, unlike other herpesviruses, its reactivation is extremely difficult both in natural and experimental conditions and has been reported very rarely. In natural infections, CpHV-1 is reactivated during the estrus phase but only in animals with low neutralizing antibody titers. In previous studies, reactivation of latent CpHV-1 has been experimentally induced in adult goats using administration of a high dose of dexamethasone for several days [8]. Interestingly, after reactivation or experimental infection, even when the virus has been inoculated intranasally, elimination via the genital route takes far longer than by the nasal route. The results of these studies indicate that CpHV-1 recognizes the genital tract as a target [8].

On goat farms, the control of CpHV-1 is based on prevention and eradication. Different types of vaccines have been investigated since the 2000s. However, vaccines for CpHV-1 have not been released as this pharmaceutical market is not economically profitable. Consequently, the control of this infection relies on hygienic prophylactic measures [10], and the research for alternative solutions is needed.

CpHV-1 has a significant biological similarity to HSV-2 considering its ability to induce latent infection in the sacral ganglia and similar genital lesions [8]. This has suggested the use of CpHV-1 infection in goat as a model for the study of HSV-2 infection in humans [11,12].

The immunosuppressive drug, mizoribine, when combined with aciclovir has been evaluated in vitro, proving effective against CpHV-1 [13]. The administration of cidofovir has also raised interest for the treatment of genital lesions in the caprine species based on in vivo and in vitro tests [12]. PHA767491, an anti-tumor drug, has been used against HSV-1, HSV-2 [14], and CpHV-1 [15]. Some natural substances, such as essential oils, have been tested for their anti-infective properties. Volatile oils of *Melissa officinalis Lamiaceae* effectively inhibited HSV-2 replication [16]. Ginger essential oil was found to have virucidal activity, inactivating CpHV-1 by up to 100% [17]. Moreover, fig latex has also shown efficacy against CpHV-1 in vivo and in vitro [11]. In addition, several essential oils have been tested against human viruses [18]. However, the use of essential oils in veterinary medical practice is limited. 

Treatment with ozone (O_3_) is an alternative therapy that uses O_3_ in a mixture with oxygen (O_2_) for medical purposes [19]. O_3_ is an allotropic form of oxygen, composed by three oxygen atoms, organized in a relatively unstable cyclic structure that makes it a powerful oxidant agent [20]. Due to this feature, it shows microbicidal and antimicrobial properties against bacteria, fungi, viruses, and protozoa [19,21,22]. As for the effects against viruses, O_3_ causes structural damage by protein and lipid peroxidation of the envelope and capsid, respectively, and by the destruction of nucleic acids [23,24]. Nucleic acid damage is evident by the disruption of specific regions of the viral genome. Some authors exposed poliovirus type 1 to ozonized water, demonstrating a specific damage in the 5′-non-coding regions of the genome [24]. Protein peroxidation plays a key role in the inactivation of non-enveloped viruses. Thurston-Enriquez et al. [25] inactivated feline calicivirus and adenovirus type 40 using ozonized water at 300 and 60 µg/L, respectively. Encouraging results have been achieved by Dubuis et al. [26] on murine norovirus and phage viruses using O_3_ in air treatment at low concentrations (0.23 ppm equal to 230 µg/L). Lipid peroxidation is the main procedure used to inactivate enveloped viruses. In a study conducted by Wells et al. [27], human immunodeficiency virus type 1 was inactivated in vitro by O_3_ in a dose-dependent manner. Severe acute respiratory syndrome coronavirus type 2 (SARS-CoV-2) viral titer on different materials (fleece, gauze, wood, glass, and plastic) significantly decreased after 30 min/2 h of exposure to O_3_ in a plexiglass chamber (0.2–4 ppm equal to 200–4000 µg/L) [28]. A gaseous mixture of 21% O_3_ in air for 80 min was able to trigger a 4-fold reduction of influenza A virus titer. Conversely, this mixture was ineffective against respiratory syncytial virus [29].

O_3_ displayed in vitro virucidal activity on HSV-1 and bovid herpesvirus type 1 (BoHV-1), inducing viral inhibition by more than 90% after 3 h of exposure [30]. Nevertheless, data regarding the virucidal efficacy of O_3_ against CpHV-1 and HSV-2 are not available.

In large animal veterinary medicine, O_3_ has been used for systemic treatment using auto-haemo [31,32] or topical [19] administration. Moreover, O_3_ has been used to treat postpartum pathologies [33] and to improve reproductive parameters in postpartum dairy cows [34,35] and to increase the fertility rates in cows affected by urovagina [36]. O_3_ therapy seemingly matched or outperformed antibiotics treatments, reducing the risks of antimicrobial resistance [34,37] and withdrawal times for meat and milk because it does not leave residue in biological tissue [37]. In goat medicine, few studies have been conducted on the application of O_3_ therapy, and they are mainly focused on reproductive [34] and milk production [38] performances.

The aim of this study was to evaluate the in vitro virucidal and antiviral effects of a medical O_3_/O_2_ gaseous mixture against CpHV-1.

## 2. Materials and Methods

### 2.1. O_3_ Generator

An O_3_ medical generator (Vet-Ozone Medica srl-Italy, Bologna, Italy) was used to produce an ozone/oxygen (O_3_/O_2_) gas mixture. After connection to an electrical source and to an O_2_ cylinder, the generator produced electrical discharges that convert O_2_ (substrate) into O_3_. The generator can produce a gas mixture containing 20 and 50 µg of O_3_/mL.

### 2.2. Hermetic Box for Gas Flow

An in-house method was developed to expose the Petri dishes to the O_3_ gas flow, as previously described [22].

Two silicon tubes were assembled on the cover of a polypropylene hermetic box. The tube for the incoming flow was connected to the O_3_ generator and the output tube to a drainpipe.

After placing the uncovered Petry dishes inside the box, the box was hermetically sealed, and the ozone generator was switched on. The generated ozonized gas mixture entered the box through tube 1. Subsequently, the gas mixture came into contact with the Petri dishes and exited through tube 2, allowing a continuous gas flow (Figure 1). The box was disinfected between each test using sodium hypochlorite (1%) with a contact time of at least 1 min as suggested in the guidelines for “Disinfection and sterilization in healthcare facilities” [39].

### 2.3. Cells and Virus

Madin–Darby bovine kidney cells (MDBK) were kindly provided by the Cell Substrate Center of the Experimental Zooprofilactic Institute of Lombardy and Emilia–Romagna. The cells were cultured at 37 °C in a 5% carbon dioxide (CO_2_) atmosphere in Dulbecco minimum essential medium (D-MEM) supplemented with 10% foetal bovine serum, 100 IU/mL penicillin, 0.1 mg/mL streptomycin, and 2 mM l-glutamine. The same medium was used for the antiviral assays. The CpHV-1 strain Ba-1, previously isolated from vaginitis in goat, was cultured and titrated in MDBK cells. The virus stock with a titer of 7.25 log_10_ tissue culture infectious dose (TCID_50_)/50 μL was stored at −80 °C and used for the experiments. The CpHV-1 viral suspension used in the experiments underwent preliminary centrifugation at 4000× *g* for 15 min to separate cellular debris.

### 2.4. Cytotoxicity Assay

A cytotoxicity assay was carried out in order to determine the conditions of cell exposure to O_3_ (O_3_ concentration in the gas mixture and exposure time) for the antiviral activity tests. Confluent 24 h monolayers of MDBK cells grown in 35 mm Petri dishes and maintained in D-MEM were exposed to the O_3_/O_2_ gas mixture containing different concentrations of O_3_ (20 and 50 μg/mL) at room temperature for 30 s (T1), 60 s (T2), 90 s (T3), 120 s (T4), 180 s (T5), and 300 s (T6). Negative controls were prepared by adding cells inside the hermetic box at the same temperature and for the same time intervals without exposure to O_3_/O_2_ gas mixture. Cytotoxicity was assessed using both direct microscopic examination of cell morphology (loss of cell monolayer, granulation, cytoplasmic vacuolization, stretching and narrowing of cell extensions, and darkening of the cell borders) [40] and indirect measurement of cell viability with an in vitro toxicology assay kit (Sigma–Aldrich Srl, Milan, Italy) based on 3-(4,5-dimethylthiazol-2 yl)-2,5-diphenyl tetrazolium bromide (XTT). The XTT test was carried out as previously described [40] by following the manufacturer’s instructions, and the obtained optical density (OD) values were used to calculate the percentage of cytotoxicity (percentage of dead cells) according to the formula: % Cytotoxicity = [(OD of control cells − OD of treated cells) × 100]/OD of control cells. The assay was performed in triplicate and the data were expressed as mean ± SD. The exposure conditions that did not reduce the viability of the treated MDBK cells by more than 20% (cytotoxicity threshold) were considered as non-cytotoxic and were selected for subsequent antiviral tests.

### 2.5. Cytophatic Effect

The cytophatic effect of CpHV-1 was evaluated on MDBK cells using an inverted microscope with live-cell imaging and hematoxylin eosin staining.

### 2.6. Virucidal Activity Assay 

The virucidal activity of O_3_/O_2_ gaseous mixture against CpHV-1 was assessed at 20 and 50 μg/mL O_3_ concentration. 

One mL of CpHV-1 stock virus was poured in a 35 mm Petri dishes and directly exposed to the O_3_/O_2_ gas mixture in the modified hermetic box at room temperature. At different time intervals (T1 to T6), 100 µL of the treated viral suspension was collected for subsequent viral titration.

A 1 mL aliquot of CpHV-1 stock virus was left untreated at room temperature and similarly sampled for viral titration, serving as virus control. The experiments were performed in triplicates.

### 2.7. Antiviral Assays

On the basis of the cytotoxicity assay results, the antiviral activity against the CpHV-1 strain Ba-1 was evaluated using the O_3_/O_2_ gaseous mixture containing O_3_ at 20 and 50 μg/mL for different exposure times (T1 and T2). To identify the step of viral inhibition by O_3_ against CpHV-1, two different protocols (A and B) were carried out as detailed below. All experiments were performed in triplicate.

#### 2.7.1. Protocol A: Virus Infection of Cell Monolayers before Treatment with O_3_

Confluent monolayers of MDBK cells of 24 h in 24-well plates were used. Cells were infected with 100 μL of viral suspension containing 100 TCID_50_ CpHV-1. After virus adsorption for 1 h at 37 °C, the viral inoculum was removed, and cell monolayers were washed once with D-MEM before adding 1 mL of maintenance medium (D-MEM). Then, cell monolayers were treated with the O_3_/O_2_ gaseous mixture. Untreated infected cells were used as virus control. After 72 h, aliquots of the supernatants were collected for subsequent viral titration. 

#### 2.7.2. Protocol B: Virus Infection of Cell Monolayers after Treatment with O_3_

Confluent monolayers of MDBK cells of 24 h in 24-well plates were used. Cells were treated with the O_3_/O_2_ gaseous mixture. Then, the monolayers were washed once with D-MEM and infected with 100 μL viral suspension containing 100 TCID_50_ CpHV-1. After virus adsorption for 1 h at 37 °C, the inoculum was removed and the monolayers were washed with D-MEM before adding 1 mL of maintenance medium (D-MEM). Untreated infected cells were used as virus control. After 72 h, aliquots of each supernatants were collected for subsequent viral titration.

### 2.8. Viral Titration

Ten-fold dilutions (up to 10^−8^) of each supernatant were titrated in quadruplicates in 96-well plates containing MDBK cells. The plates were incubated for 72 h at 37 °C in 5% CO_2_. Cytopathic effect of CpHV-1 on MDBK cells was evaluated using an inverted microscope with live-cell imaging or using haematoxylin eosin staining. Based on the cytopathic effect, TCID_50_/50 μL was calculated by following the Reed–Muench method [41]

### 2.9. Data Analysis

All data were expressed as mean ± SD and analyzed using GraphPad Prism (v 9.5.0) program (Intuitive Software for Science, San Diego, CA, USA). To assess the normality of distribution, Shapiro–Wilk test was performed. Two-way factorial ANOVA, with concentration * time as factors, and Tukey test as post hoc test were applied to cytotoxicity results. Student’s *t* tests for independent samples were performed on virucidal and antiviral activity results (*p* < 0.05).

## 3. Results

### 3.1. Cytotoxicity Assay

Direct exposure of MDBK cells to O_3_/O_2_ gas mixture containing O_3_ at 20 and 50 μg/mL did not produce any changes in cell morphology at T1 and T2, whereas morphological signs of cytotoxicity were consistently observed in cells exposed to O_3_ at 20 and 50 μg/mL for longer time intervals (i.e., at T3 to T6).

Morphological observations overlapped indirect measurements of cytotoxicity using the XTT test. Cell exposure to O_3_ at 20 and 50 μg/mL at different time intervals (T1 to T6) resulted in increasing cytotoxicity in a dose- and time contact-dependent fashion (Figure 2). O_3_ at 20 μg/mL at T1 and T2 induced cytotoxicity of 0.53% (±0.15) and 3.64% (±0.8), respectively, which was below the cytotoxic threshold. Higher cytotoxicity of 31.03% (±1.1), 36.78% (±1.2), 40.10% (±1.3), and 81.52% (±2.3) was observed at T3, T4, T5, and T6, respectively (Figure 2A). 

O_3_ at 50 μg/mL at T1 and T2 produced cytotoxicity of 0.51% (±0.13) and 3.61 % (±0.95), respectively, which was below the cytotoxic threshold. Higher cytotoxicity of 59.77% (±1.3), 65.51% (±1.6), 82.57% (±1.8), and 85.39% (±2.6) was observed at T3, T4, T5, and T6, respectively (Figure 2B).

The ANOVA model showed a statistically significant decrease in cytotoxicity in MDBK cells treated with O_3_ at 20 (F = 1517, *p* < 0.0001) and 50 (F = 1822, *p* < 0.0001) μg/mL between different time intervals (T1–T6). Using a two-by-two comparison of cytotoxicity induced by O_3_ at 20 and 50 μg/mL, a statistically significant decrease in cytotoxicity was consistently observed at different time intervals (T1–T6). Conversely, the comparison between O_3_ at 20 μg/mL at T4 and T5 and between O_3_ at 50 μg/mL at T5 and T6 lacked statistical significance (*p* > 0.05). 

On the basis of these results, the antiviral activity assays were carried out using O_3_ at 20 and 50 μg/mL at T1 and T2, which were below the cytotoxicity threshold.

### 3.2. Cytophatic Effect

Cytopathic effect of CpHV-1 on MDBK cells is displayed in Figure 3.

### 3.3. Virucidal Activity Assay

Data obtained were analyzed using Shapiro–Wilk test, confirming the normality of distribution (W = 0.8137, *p* > 0.05). Data from the virucidal activity assay showed that the O_3_/O_2_ gaseous mixture containing O_3_ at 20 μg/mL significantly reduced CpHV-1 titer by 1.25 log_10_ TCID_50_/50 μL (*p* < 0.05) at T1 and T2, 1.50 log_10_ TCID_50_/50 μL (*p* < 0.05) at T3 to T5, and 2.00 log_10_ TCID_50_/50 μL at T6 (*p* < 0.0001) when compared with the untreated control.

Data from the virucidal activity assay showed that the O_3_/O_2_ gas mixture containing O_3_ at 50 μg/mL significantly reduced CpHV-1 titer by 1.25 log_10_ TCID_50_/50 μL (*p* < 0.05) at T1 and T2, 1.50 log_10_ TCID_50_/50 μL (*p* < 0.05) at T3 to T4, 1,75 log_10_ TCID_50_/50 μL (*p* < 0.05) at T5, and 2.00 log_10_ TCID_50_/50 μL at T6 (*p* < 0.0001) when compared with the untreated control (Figure 4). 

### 3.4. Antiviral Assays

#### 3.4.1. Protocol A: Virus Infection of Cell Monolayers before Treatment with O_3_

Comparing in terms of viral titer the untreated infected cells (7.25 ±0.25 log_10_ TCID_50_/50 μL) with the infected cells treated with the O_3_/O_2_ gas mixture containing O_3_ at 20 μg/mL at T1 and T2 (7.00 ± 0.25 log_10_ TCID_50_/50 μL), a slight decrease in viral titer (0.25 log_10_) was observed, however, with no statistical significance (*p* > 0.05). Comparing the viral titer of the untreated infected cells (7.25 ± 0.25 log_10_ TCID_50_/50 μL) and of the infected cells treated with O_3_ at 50 μg/mL at T1 and T2 (6.00 ±0.25 log_10_ TCID_50_/50 μL), a significant decrease in the viral titer (1.25 log_10_) was observed (*p* < 0.05) (Figure 5). 

#### 3.4.2. Protocol B: Virus Infection of Cell Monolayers after Treatment with O_3_

Comparing the viral titer of the untreated infected cells (7.25 ± 0.25 log_10_ TCID_50_/50 μL) with the viral titer of the infected cells pre-treated with the O_3_/O_2_ gas mixture containing O_3_ at 20 and 50 μg/mL at T1 and T2 (7.25 ± 0.25 log_10_ TCID_50_/50 μL), no decrease in viral titer was observed (Figure 6).

## 4. Discussion

O_3_ therapy is largely used in veterinary medicine for its disinfectant, anti-inflammatory, immunostimulant, and antimicrobial effects [19].

In this study, we have focused on the activity of ozone therapy against the genital herpesvirus of goat (CpHV-1) to decipher a possible field application in veterinary and human medicine. Indeed, CpHV-1 and HSV-2 share important biological characteristics, and the infection by CpHV-1 in goats is considered a valid animal model for the study of infection of HSV-2 and its therapy in humans, [15].

There are several in vivo and in vitro studies published in the literature, addressing/demonstrating the therapeutic potential of O_3_ in treating genital infections of farm animals [22,35] and documenting the disinfectant, immunomodulatory, and anti-inflammatory actions of O_3_. Inoculation of O_3_ using foams into the vagina and uterus of cows affected by urovagina has been shown to decrease the “open days” and the number of artificial inseminations required to induce pregnancy in cows affected by urovagina. Moreover, the beneficial role of O_3_ in the repair process of the vaginal and cervical mucosa was observed [36]. The interest of clinical researchers in new therapies, such as ozone therapy, aid future studies aimed at the treatment of infectious pathologies that require the use of antibiotics.

The virucidal effect of O_3_ has been reported for different viruses. This gas has a potent oxidant action on microorganisms [23,24,25,26,27,28,29], damaging the lipidic envelope and protein capsid of viruses [23]. In addition, O_3_ could inactivate viruses by destroying guanine residues of nucleic acids [42] as demonstrated for poliovirus type 1 [24,43]. 

In this study, an O_3_/O_2_ gas mixture containing O_3_ at 20 and 50 μg/mL was evaluated against CpHV-1 at different time points (T1 to T6) to assess its virucidal properties. Furthermore, the in vitro antiviral activity of the O_3_/O_2_ gas mixture containing O_3_ at 20 and 50 μg/mL against CpHV-1 was evaluated at two different time points, T1 and T2. The concentrations of 20 and 50 μg/mL were chosen based on the cytotoxic activity obtained using XTT test on MDBK cells for different time points (T1–T6). Both O_3_ concentrations were regarded as non-cytotoxic (below the cytotoxicity threshold of 20%) at T1 and T2. At later time points, starting from T3, an increase in cytotoxicity was observed primarily at the concentration of 50 μg/mL (over 60%).

In other studies, concentrations from 10 to 20 μg/mL of O_3_ in O_3_/O_2_ gas mixture (generated with a medical ozone generator as in our study) were assessed in other cell lines, i.e., HeLa [44] and SH-SY5Y cells (a human neuroblastoma cell line), and did not display cytotoxic effect [45]. These concentrations did not induce significant alterations in cell viability, and cellular mortality was observed only when cells were treated with O_3_ at 100 μg/mL [45].

Eukaryotic cells demonstrate a certain resistance in vitro to the prooxidant effect of O_3_ because they are protected by the presence of albumin, which with its reducing group -SH is one of the most protective compounds [46]. Of course, the adopted O_3_ concentration is crucial as high concentrations could overwhelm this protective mechanism, leading to cell damage and death [47].

In the virucidal activity assay, exposure of CpHV-1 to the gas mixture was able to reduce significantly the viral titer in a time-dependent manner, leading to a decrease in viral titer of up to 2.00 log_10_ TCID_50_/50 μL at T6.

To evaluate the antiviral activity at the maximum non-cytotoxic dose of O_3_ at 20 and 50 μg/mL at T1 and T2, in order to identify the phase in which viral replication might be inhibited, cells were infected with CpHV-1 before (protocol A) and after (protocol B) the treatment with O_3_.

In protocol A, when O_3_ was used at a concentration of 20 μg/mL, we observed a very slight and statistically insignificant reduction in viral titer (0.25 log_10_ TCID_50_/50 μL), suggesting that O_3_ was not able to inhibit virus replication. O_3_ at a 50 μg/mL concentration induced a statistically significant reduction of viral titer (1.25 log_10_ TCID_50_/50 μL). 

Pre-treatment of the cells with O_3_ at 20 and 50 μg/mL (protocol B) did not reduce the viral titer, hinting a lack of inhibition of O_3_ in virus uptake and replication.

Overall, as significant results were obtained with short exposure times, the use of O_3_ in vivo could be implemented, primarily in the veterinary field. Future studies could address the use of O_3_ in CpHV-1-infected goats to gain more translational information for human herpesvirus genital infection. In a previous report, the inactivation of herpes viruses (HSV-1 and BoHV-1) with O_3_ was achieved by applying a long exposure time (1 to 3 h) [30]. Compared to other studies [15,17,28,29,30,31,32], the contact time of the O_3_/O_2_ gas mixture required to trigger significant effects against CpHV-1 was lower, and this could be an advantage for in vivo experiments. Long treatment times would not be ideal due to excessive stress induced to animals, mainly for ones in animal containment.

## 5. Conclusions

We reported the in vitro virucidal and antiviral activity of a medical O_3_/O_2_ gaseous mixture against CpHV-1. A short exposure of the virus to O_3_ at low concentration (20 μg /mL) was required to achieve partial virus inactivation. This study represents the first step to assess the clinical efficacy of O_3_ therapy for the treatment of genital herpes infection. Further essential steps will be the evaluation of the in vitro effects on vulvar and vaginal epithelial cells as well as of the efficacy of treatment of CpHV-1-associated genital lesions in infected goats in vivo. Furthermore, it might be interesting to test whether O_3_ is also effective on HSV-2 given the close biological similarity with CpHV-1.

## Figures and Tables

**Figure 1 animals-13-01920-f001:**
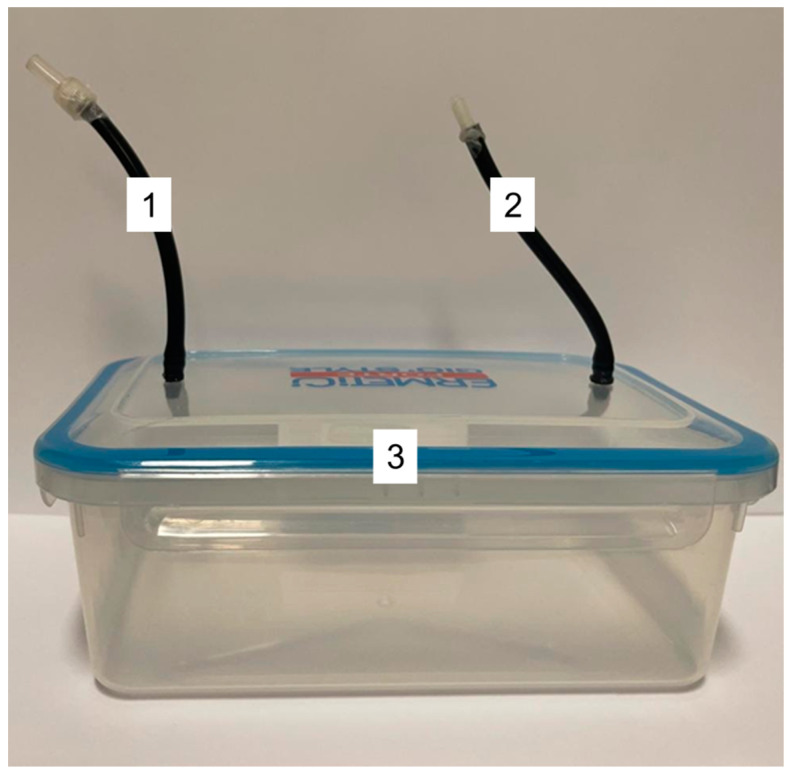
Modified hermetic box for continuous gas flow. The device is composed of two silicone tubes (one tube for gas entry (1) and one tube for gas exit (2)) and of a polypropylene hermetic box (3).

**Figure 2 animals-13-01920-f002:**
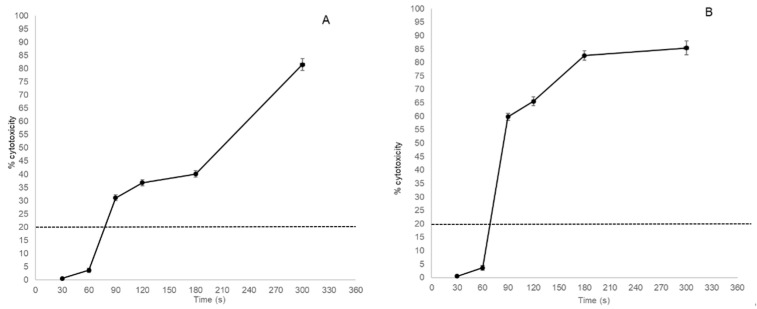
Cytotoxicity of MDBK cells treated with O_3_/O_2_ gas mixture containing O_3_ at 20 μg/mL (**A**) and 50 μg/mL (**B**) plotted against time of exposure. The horizontal dotted line indicates the threshold of cytotoxicity (20% of cell death).

**Figure 3 animals-13-01920-f003:**
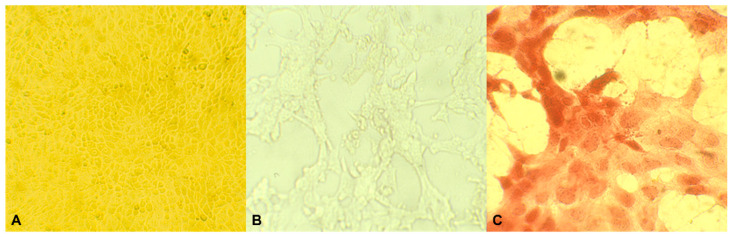
Twenty four-hour monolayer of Madin–Darby bovine kidney (MDBK) cells (magnification 10×) (**A**); Cytopathic effect of CpHV-1 on MDBK cells with live-cell imaging (magnification 40×) (**B**); Cytopathic effect of CpHV-1 on MDBK cells using hematoxylin-eosin staining (magnification 40×) (**C**).

**Figure 4 animals-13-01920-f004:**
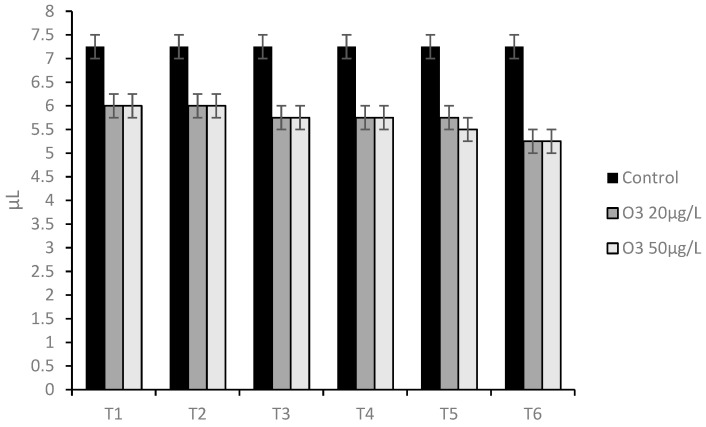
Viral titration on Madin–Darby bovine kidney (MDBK) cells inoculated with caprine herpes virus 1 (CpHV-1) and not treated (Control) or treated with Ozone/Oxygen (O_3_/O_2_ 20 and 50 μg/mL) at room temperature for 30 s (T1), 60 s (T2), 90 s (T3), 120 s (T4), 180 s (T5), and 300 s (T6).

**Figure 5 animals-13-01920-f005:**
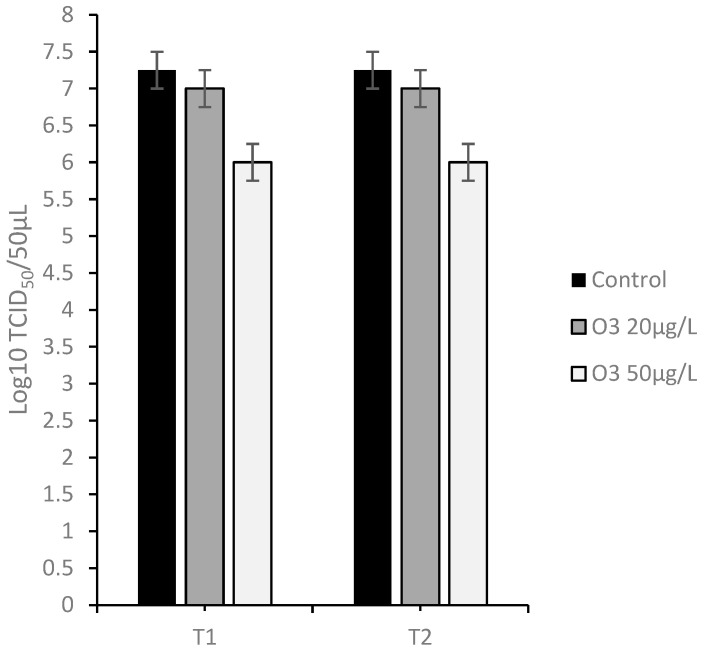
Viral titration on Madin–Darby Bovine Kidney (MDBK) cells inoculated with Caprine herpes virus 1 (CpHV-1), treated after virus inoculation with Ozone/Oxygen (O_3_/O_2_ 20 and 50 μg/mL) at room temperature for 30 s (T1), 60 s (T2), and untreated cells (Control).

**Figure 6 animals-13-01920-f006:**
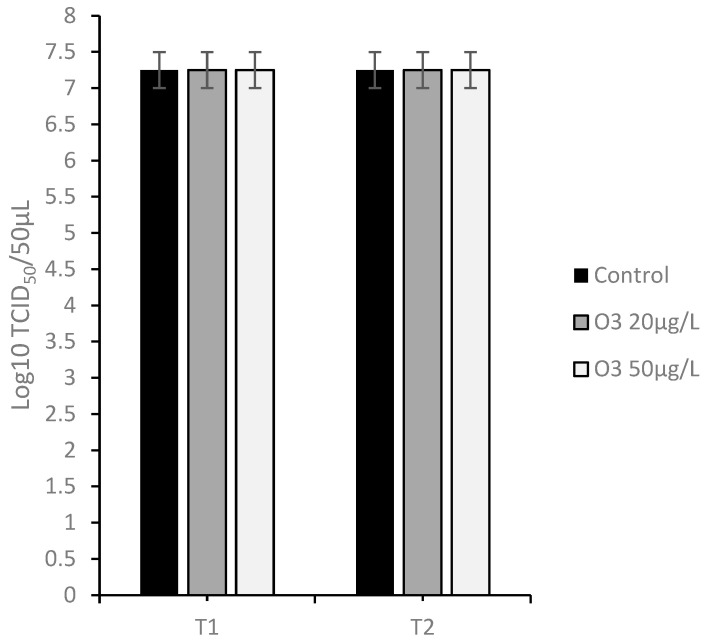
Viral titration on Madin–Darby bovine kidney (MDBK) cells inoculated with caprine herpes virus 1 (CpHV-1) treated before inoculation with ozone/oxygen (O_3_/O_2_ 20 and 50 μg/mL) at room temperature for 30 s (T1), 60 s (T2) and untreated cells (Control).

## Data Availability

Data is available within the article.

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
