# Peer review of "In Vitro Activity of Ozone/Oxygen Gaseous Mixture against a Caprine Herpesvirus Type 1 Strain Isolated from a Goat with Vaginitis"

_animals, 2023, doi:10.3390/ani13121920_

Round 1
Reviewer 1 Report (Previous Reviewer 2)
No further comments
Some minor spelling errors and phrasing could be corrected.
Author Response
Thank You for your suggestion. We revised the manuscript and corrected some errors.
Reviewer 2 Report (Previous Reviewer 1)
All comments have been addressed and the manuscript has been improved.
Author Response
Thank You
Reviewer 3 Report (New Reviewer)
In vitro virucidal activity of an O3/O2 gas mixture containing O3 at 20 and 50 μg/mL, against CpHV-1 was evaluated at different time points (T1 to T6). Moreover, the Authors studied the in vitro antiviral activity of an O3/O2 gas mixture containing O3 at 20 and 50 μg/mL, against CpHV-1. Encouraging results have been obtained to be applied in both veterinary and human medicine in the eventual fight against Herpes virus infections. Data are supported by statistical analysis.
- Line 20: Please uniform the word ‘titre’ in the abstract and manuscript
- Line 21: please ‘in vivo’ in italics
- Line 262: please uniform the p-value
- Figures 2A and 2B are not clearly visible in the page
Author Response
Thank You for your comments. We took in account all your suggestions.
This manuscript is a resubmission of an earlier submission. The following is a list of the peer review reports and author responses from that submission.
Round 1
Reviewer 1 Report
1- The authors should test the cytotoxic effect of the ozone\oxygen gas mixture on the cultured cells used in viral titration (MDBK cells) to confirm that this reduction in the viral titre is related to the antiviral effect of the gas mixture.
2- The authors mentioned in the discussion section that concentration and exposure times were chosen because they have already been evaluated as effective in inhibiting the bacterial growth of metritis-causing bacteria in cattle. I disagree with the authors in this point because the viral work is completely different than bacterial work and they have to test at least different concentrations of the gas mixture on the virus.
3- The authors should clarify how did they calculated the TCID50 in the viral titration assay in M&M section.
4- Since T4 was the exposure time that gave 2 fold reduction, So why do not authors try to test longer exposure time to see if getting higher reduction.
Author Response
To the Editor of Animals
Dear Editor
I send you the revised manuscript: “Oxygen/ozone gaseous mixture activity in vitro on Caprine Herpesvirus type 1 (CpHV-1)” by Lillo et al.
We have found the Reviewers’ comments very helpful and we believe we have addressed each question made by the Reviewers in the revised manuscript enclosed here. Please find below a point-by-point reply to each reviewer.
Best regards,
Annalisa Rizzo
Valenzano (BA), 22.01.2023
Prof. Rizzo DVM, PhD, Full professor
Department of Veterinary Medicine, University of Bari, Valenzano (BA), Italy
Email: annalisa.rizzo@uniba.it
Answers to Reviewers’ comments and suggestions for authors
Reviewer 1
R1Q1: The authors should test the cytotoxic effect of the ozone\oxygen gas mixture on the cultured cells used in viral titration (MDBK cells) to confirm that this reduction in the viral titre is related to the antiviral effect of the gas mixture.
R1A1: Cytotoxicity on MDBK cells was assessed relying on two different approaches: directly exposing the gas mixture on the 24h monolayers of MDBK cells for different contact times or by supplementing 24-h monolayers of MDBK cells with D-MEM pre-treated with the gas mixture for different contact times. This information was added in Materials and Methods and Results sections at page 4, lines 124-131 and 157-160 and page 5 lines 159-167.
R1Q2: The authors mentioned in the discussion section that concentration and exposure times were chosen because they have already been evaluated as effective in inhibiting the bacterial growth of metritis-causing bacteria in cattle. I disagree with the authors in this point because the viral work is completely different than bacterial work and they have to test at least different concentrations of the gas mixture on the virus.
R1A2: We agree with the Referee’s comment that the rational of our study design is not clear. Since the Ozone device used in this study can only erogate Ozone at limited concentrations (20, 35, 50 μg/ml), this was a constraint of our study design. We decided to use 20 μg/ml concentration as it was the lowest concentration used by the instrument, and to avoid cytotoxic effects on cells. Concentrations higher than 50 μg of Ozone/mL can induce cell death (Costanzo et al., 2015; Scassellati et al., 2017). However, as requested by the Referee, we performed another experiment using O3/O2 gas mixture containing O3 at 50 μg/mL and we observed that the virucidal activity at this higher concentration was comparable with that induced by O3/O2 gas mixture containing O3 at 20 μg/mL. Accordingly, we did not include these results in the manuscript.
References
Costanzo, M., Cisterna, B., Vella, A., Cestari, T., Covi, V., Tabaracci, G., Malatesta, M., 2015. Low ozone concentrations stimulate cytoskeletal organization, mitochondrial activity and nuclear transcription. Eur. J. Histochem. 59, 129–136. https://doi.org/10.4081/ejh.2015.2515
Scassellati, C., Costanzo, M., Cisterna, B., Nodari, A., Galiè, M., Cattaneo, A., Covi, V., Tabaracci, G., Bonvicini, C., Malatesta, M., 2017. Effects of mild ozonisation on gene expression and nuclear domains organization in vitro. Toxicol. Vitr. 44, 100–110. https://doi.org/10.1016/j.tiv.2017.06.021
R1Q3: The authors should clarify how did they calculated the TCID50 in the viral titration assay in M&M section.
R1A3: We added this information in the text suggested (page 4, lines 147-151)
R1Q4: Since T4 was the exposure time that gave 2-fold reduction, so why do not authors try to test longer exposure time to see if getting higher reduction.
R1A4: we agree with Refeere 1’s comment. Unlike our study, in previous reports relying on Ozone virucidal activity longer times of exposure (even some hours) were evaluated using Ozone generators specific for room disinfection (Steinmann et al., 2021). The Ozone generator used in our study is a medical device and the in vitro results obtained in this study open perspectives in terms of in vivo treatment of goat genital herpetic lesions which can be used as a possible model for human genital infection. The choice of exposure times derives from veterinary practice as it may be impractical to treat goats for times longer than those used in this study due to the exceeding stress induced to the animals. This information were detailed in the tex at page 6 lines 226-228.
References
Steinmann, J., Burkard, T., Becker, B., Paulmann, D., Todt, D., Bischoff, B., Steinmann, E., Brill, F.H.H., 2021. Virucidal efficacy of an ozone-generating system for automated room disinfection. J. Hosp. Infect. 116, 16–20. https://doi.org/10.1016/j.jhin.2021.06.004
Reviewer 2 Report
In this paper, the authors have described studies on the action of ozone on a caprine herpesvirus as a possible virucidal agent that could be used to treat herpesvirus infections in goats. At its present state of development, the work presented is very preliminary. More work to substantiate loss of infectivity needs to be provided to form the basis of the authors' conclusion that ozone is virucidal for caprine herpesvirus.
Major Points
1. Data should have been provided to show the images from the control and infected MDCK cells to allow us to judge the conclusions i.e. light micrographs of the cells. This should also be supplemented by staining, e.g. by crystal violet to show reduction in infectivity following ozone treatment of the virus.
2. Another method should be provided to show reduction in infectivity - as it stands, we are not shown enough data to make a firm conclusion based on CPE. For example, Q-PCR analysis of viral DNA in cells infected with control and ozone-treated virus or, if an antibody is available against any viral protein, a loss of antigen expression in cells exposed to ozone-treated virus by IF or Western blot.
3. More details need to be given on how the virus was purified and it's state of purity when ozone-treated. Was it gradient-purified or was it used as a crude cell extract of infected MDCK cells?
Minor Point
Figure 1 and 2 seem unnecessary since Fig 1 deals with a commercially-available piece of equipment and Fig 2 seems obvious.
Author Response
Reviewer 2
In this paper, the authors have described studies on the action of ozone on a caprine herpesvirus as a possible virucidal agent that could be used to treat herpesvirus infections in goats. At its present state of development, the work presented is very preliminary. More work to substantiate loss of infectivity needs to be provided to form the basis of the authors' conclusion that ozone is virucidal for caprine herpesvirus.
Major Points
R2Q1: Data should have been provided to show the images from the control and infected MDBK cells to allow us to judge the conclusions i.e. light micrographs of the cells. This should also be supplemented by staining, e.g. by crystal violet to show reduction in infectivity following ozone treatment of the virus.
R2A1: As suggested by the referee R2, we added pictures showing the uninfected and infected cell monolayers, and their visualizations by live-cell imaging and hematoxylin eosin staining.
R2Q2: Another method should be provided to show reduction in infectivity - as it stands, we are not shown enough data to make a firm conclusion based on CPE. For example, Q-PCR analysis of viral DNA in cells infected with control and ozone-treated virus or, if an antibody is available against any viral protein, a loss of antigen expression in cells exposed to ozone-treated virus by IF or Western blot.
R2A2: We agree with the referee that qPCR is a good proxy to assess the extent of replication of the virus and, in several cases, we couple molecular quantification (qPCR) with virus titration. However, since qPCR cannot distinguish between infectious and noninfectious viral particles, virus titration in cells is the elective option to assess residual infectivity. In the case of non-cultivatable viruses, surrogate strategies can be implemented, but CpHV is easily cultivatable in MDBK cells.
R2Q3: More details need to be given on how the virus was purified and its state of purity when ozone-treated. Was it gradient-purified or was it used as a crude cell extract of infected MDCK cells?
R2A3: the CpHV-1 viral suspension used in the experiment undergo a preventive centrifugation at 4000 xg for 15 min to separate cellular debris (page 4, lines 134-135).
Minor Point
R2Q4: Figure 1 and 2 seem unnecessary since Fig 1 deals with a commercially-available piece of equipment and Fig 2 seems obvious.
R2A4: We removed Figure 1 as suggested. We would like to retain figure 2 (figure 1 in the revised manuscript), since it depicts the technical modalities used to expose the virus and cells to the gas flow. We think this could be of interest for some readers. Yet, we agree that this could be obvious for others.
Round 2
Reviewer 1 Report
It seems that the authors were in a rush in addressing some points, especially the following;
IN R1A1; The authors have mentioned that they have performed the cytotoxicity assay according to (Lanave et al., 2017) and determined the cytotoxicity directly by microscopic examination of cell morphology (Lines 158-161).
1- The authors forgot to put this reference in the reference list.
2- If the authors went through their paper (Lanave et al., 2017), they will find that they assessed the cytotoxicity using the In Vitro Toxicology Assay Kit and performed XTT cell proliferation assay. How can the authors cite a reference for an experiment which they have not performed in their current manuscript?!!
3- How can it be possible scientifically to assess the cytotoxicity of such gases which they claim that might be promising therapeutics for HSV-2 in humans through just a microscopic examination of the cell morphology?!!!
Author Response
Dear Editor
I send you the revised manuscript: “Oxygen/ozone gaseous mixture activity in vitro on Caprine Herpesvirus type 1 (CpHV-1)” by Lillo et al.
We lengthened the text and modified the similarities to other articles. Moreover we have found the Reviewer comments very helpful and we have answered to each question made by the Reviewer in the revised manuscript enclosed here. Please find below a point-by-point reply.
Dear Reviewer,
thank You for your comments. Following You can find the answer to your comments.
It seems that the authors were in a rush in addressing some points, especially the following;
IN R1A1; The authors have mentioned that they have performed the cytotoxicity assay according to (Lanave et al., 2017) and determined the cytotoxicity directly by microscopic examination of cell morphology (Lines 158-161).
R1Q1- The authors forgot to put this reference in the reference list.
R1A1: Sorry, we added it.
R1Q2- If the authors went through their paper (Lanave et al., 2017), they will find that they assessed the cytotoxicity using the In Vitro Toxicology Assay Kit and performed XTT cell proliferation assay. How can the authors cite a reference for an experiment which they have not performed in their current manuscript?!!
R1Q3- How can it be possible scientifically to assess the cytotoxicity of such gases which they claim that might be promising therapeutics for HSV-2 in humans through just a microscopic examination of the cell morphology?!!!
R1A2/3: We agree with the referee's observations. In the cited reference (Lanave et al., 2017), cytotoxicity test has been performed through microscope observation and XTT test. XTT test was pivotal in that study as antiviral activity, i.e. the inhibition of viral replication, was evaluated in vitro.
Conversely, in our study, the evidence was based on virucidal activity. In detail, ozone was posed directly in contact with the virus and then it was transferred to MDBK cells to evaluate its viability.
We kindly ask that our explanation could be accepted by the referee.
Otherwise we would ask for an two-weeks extension of the deadline to accomplish the cytotoxicity test that would be useful for the in vivo study.
Best regards,
Annalisa Rizzo
Reviewer 2 Report
No further comments
Author Response
Thank You